

# Behavioral dominance interactions between *Nicrophorus orbicollis* and *N. tomentosus* burying beetles (Coleoptera: Silphidae)

Scott D. Schrempf, Kevin W. Burke, Jillian D. Wettlaufer and Paul R. Martin

Department of Biology, Queen's University, Kingston, ON, Canada

## ABSTRACT

Asymmetric interference competition, where one species is behaviorally dominant over another, appears widespread in nature with the potential to structure ecological communities through trade-offs between competitive dominance and environmental tolerance. The details of how species interact and the factors that contribute to behavioral dominance, however, are poorly known for most species, yet such details are important for understanding when and why trade-offs occur. Here, we examine behavioral interactions between two species of burying beetles (Coleoptera: Silphidae) that compete for limited breeding resources (i.e., small vertebrate carcasses) in nature, to identify behaviors involved in interference competition and to test if large body size, species identity, or time of arrival best predict behavioral dominance among species. To test these ideas, we placed same-sex individuals of *Nicrophorus orbicollis* (early to mid-summer breeder) and *N. tomentosus* (late summer to fall breeder) into an enclosure together with a 25–30 g mouse carcass (*Mus musculus*). We then video-recorded all behaviors, including neutral and aggressive interactions, for 13 h per trial ($N = 14$ trials). For each interaction, we assigned a winner based on which beetle retained its position instead of fleeing or retained possession of the carcass; the overall behavioral dominant was determined as the individual that won the most interactions over the length of the trial. We found that large body size was the best predictor of behavioral dominance. In most interactions, *N. orbicollis* was larger and dominant over *N. tomentosus*; however, when *N. tomentosus* was larger they outcompeted smaller *N. orbicollis*, illustrating the importance of body size in aggressive contests. The order of arrival to the carcass (priority effects) did not predict behavioral dominance. The larger size and abundance of *N. orbicollis* in nature suggest a competitive asymmetry between the species, supporting the idea that *N. orbicollis* constrains the ability of *N. tomentosus* to breed earlier in the summer.

Corresponding author
Scott D. Schrempf,
14sds8@queensu.ca

## INTRODUCTION

Closely related species commonly share ecological traits and preferences leading to competition for resources necessary for survival and reproduction (*Violle et al., 2011*). For many closely related species, competitive interactions involve direct interference and aggression (*Peiman & Robinson, 2010*; *Grether et al., 2017*; *Drury, Cowen & Grether, 2020*), with one species behaviorally dominant over the other (*Morse, 1974*; *Martin, Freshwater & Ghalambor, 2017*). While subordinate species typically lose aggressive contests with dominants, they often persist and even thrive because they are uniquely adapted to environmental conditions to which the behaviorally dominant species is not adapted (*Morse, 1974*; *Martin, 2015*), suggesting a trade-off between behavioral dominance and environmental tolerance (*Bestelmeyer, 2000*). Understanding if and when these trade-offs occur requires an understanding of the traits that confer dominance in aggressive contests, and how these traits might compromise species' ability to tolerate different environmental conditions (*Martin, 2015*).

Burying beetles (genus *Nicrophorus*) are an excellent system for studying and understanding the traits involved in aggressive interactions and behavioral dominance. Seven species of *Nicrophorus* have been recorded in southern Ontario, with all species requiring small vertebrate carcasses for reproduction (*Anderson & Peck, 1985*). Interspecific competition for carcasses appears common, with individuals from different species aggressively competing for the same carcass (*Wilson, Knollenburg & Fudge, 1984*; *Trumbo, 1990a*; *Trumbo & Bloch, 2002*). Carcasses are usually not shared between breeding pairs, with carcass takeovers resulting in expulsion of prior residents (*Wilson, Knollenburg & Fudge, 1984*; *Trumbo, 1990a*). Adults care for their eggs and young and defend them against conspecifics and heterospecifics (*Eggert & Müller, 1997*; *Scott, 1998*). Despite competing for shared reproductive resources, *Nicrophorus* species are thought to coexist by differing in their ecological strategies that might allow for coexistence; in particular, species appear to partition resource use across space (e.g., by using different habitats; *Anderson, 1982*; *Anderson & Peck, 1985*; *Beninger, 1994*; *Lingafelter, 1995*; *Ohkawarai, Suzuki & Katakura, 1998*; *Scott, 1998*; *Wilhelm, Larson & Storey, 2001*; *Ulyshen, Hanula & Horn, 2007*; *LeGros & Beresford, 2010*; *Dyer & Price, 2013*; *Brown & Beresford, 2016*; *Wettlaufer et al., 2018*; *Keller, Howard & Hall, 2019*; *Burke et al., 2020*) and time (e.g., different seasonal patterns of activity and breeding; *Easton, 1979*; *Anderson, 1982*; *Wilson, Knollenburg & Fudge, 1984*; *Anderson & Peck, 1985*; *Trumbo, 1990b*; *Beninger, 1994*; *Lingafelter, 1995*; *Scott, 1998*; *Wettlaufer, 2019*), and some through the use of carcasses of different size (*Easton, 1979*; *Trumbo, 1990a*).

The competitive ability of *Nicrophorus* varies between species (*Pukowski, 1933*; *Easton, 1979*; *Wilson, Knollenburg & Fudge, 1984*; *Otronen, 1988*; *Trumbo, 1990a*; *Scott, 1998*) and evidence suggests that specific traits influence the outcome of aggressive contests. Body size is perhaps the best predictor of behavioral dominance among *Nicrophorus* species, with larger individuals consistently displacing smaller individuals from carcasses (*Pukowski, 1933*; *Easton, 1979*; *Wilson, Knollenburg & Fudge, 1984*; *Otronen, 1988*;

*Trumbo, 1990a*; *Scott, 1998*). When competing individuals of some species (e.g., *N. vespilloides*, *N. investigator*, and *N. vespillo*) are the same size, however, individuals of the species that is typically larger have been found to be more likely to win aggressive contests and secure a carcass (*Otronen, 1988*), suggesting that species identity or traits unrelated to body size may also influence the behavioral dominance relationships of some species. Similarly, arriving first to a carcass (priority effects) can also affect resource retention and the ability of beetles to win aggressive contests (e.g., within *N. vespilloides*; *Otronen, 1988*); however, later arriving dominant species frequently usurp carcasses from some subordinate species (*Wilson, Knollenburg & Fudge, 1984*), killing the offspring of subordinates in the process (*Trumbo, 1990a*).

Here, we examine the role of body size, species identity, and priority effects in determining behavioral dominance among two common species of *Nicrophorus* that co-occur in southeastern Ontario, Canada. *Nicrophorus orbicollis* and *N. tomentosus* overlap in breeding periods and habitat use, and are frequently captured together in baited traps at our study site, suggesting they likely compete for carcasses in nature (*Wettlaufer, 2019*; *Burke et al., 2020*). On average, *N. orbicollis* is the largest species of *Nicrophorus* at our site (*Collard et al., 2020*) and is reproductively active from early June to September, peaking in abundance from mid-June to late August (*Anderson & Peck, 1985*; *Wettlaufer, 2019*). In contrast, *N. tomentosus* is a relatively small species (*Collard et al., 2020*) that is active from July into October, peaking in abundance from late July to mid-September (*Wettlaufer, 2019*). *N. tomentosus* is the only *Nicrophorus* species at our study site known to overwinter as prepupae (*Anderson & Peck, 1985*). The ability of *N. tomentosus* to winter as prepupae may shorten the time required for development before winter diapause, uniquely allowing this species to breed later in the summer and fall at our study site (*Wilson, Knollenburg & Fudge, 1984*; all other *Nicrophorus* at our site winter as adults; *Anderson & Peck, 1985*). When searching for carrion, *N. orbicollis* is generally nocturnal while *N. tomentosus* is active during the day (*Wilson, Knollenburg & Fudge, 1984*; *Anderson & Peck, 1985*). Once a carcass has been located and secured, both species will actively prepare the carcass for breeding and defend it day and night (authors, unpublished observations; *Pukowski, 1933*; *Otronen, 1988*). Carcass preparation includes burying the carcass, removing hair, fur, or feathers, rolling the carcass into a ball, and applying oral and anal secretions to prevent decay (*Eggert & Müller, 1997*; *Scott, 1998*). While some species of *Nicrophorus* are known to parasitize broods of other species (e.g., *N. pustulatus*; *Trumbo, 1994*), interspecific brood parasitism has not been described from *N. orbicollis* or *N. tomentosus*.

Here, we use captive, same-sex arena experiments to examine how individual *N. orbicollis* and *N. tomentosus* interact when in the presence of an essential reproductive resource (mouse carcass), and whether body size, species identity, or the order of arrival predict behavioral dominance among these species. By reviewing video-recordings of interactions filmed continuously over 13-h periods, we summarize the outcomes and details of encounters among species and assign behavioral dominance based on the summed outcomes of interactions.

## MATERIALS AND METHODS

### Species collection and identification

We collected live *N. orbicollis* and *N. tomentosus* during July and August 2018 using non-lethal pitfall traps set on the properties of the Queen's University Biological Station (44.5653°N, 76.3242°W, 115–170 m elevation) near Elgin, Ontario. We identified all *Nicrophorus* beetles to species and sex following *Anderson & Peck (1985)* and placed each beetle individually into 4oz clear plastic containers (Reditainer, model no. RTSC100400) containing moist soil and ventilated with small airholes, with their species and sex marked on the lid. All beetles were housed outdoors in shade with natural light and temperature conditions until used in our trials; individuals were held in captivity 4–14 days before trials began. Beetles held for more than 1 week were fed ground beef every 7 days. All beetles were given water every 5 days, at which time the soil was also moistened.

### Captive experiments

Trials involved single *N. orbicollis* and *N. tomentosus* of the same sex competing for one mouse (*Mus musculus*) carcass. Mouse carcasses were donated by the Queen's University Health Sciences breeding facilities. The mice were culled using carbon dioxide gas and immediately frozen at −20 °C. For our trials, we used mice weighing 25–30 g, and thawed carcasses for 48 h at room temperature before experiments. We randomly selected burying beetles of the same sex representing our two focal species based on the lid labels, without regard to size, but within the same capture cohort, such that two individuals within a trial were originally captured within a day of each other. Nonetheless, we did not know the nutritional condition of our beetles or their feeding history prior to capture, and differences among individuals in condition could have affected our results.

Each pair of beetles was placed into a clear plastic container (31.4 × 20.8 × 11.9 cm) with 2 cm of fresh top soil and a mouse carcass placed in the center. Inside the container (fastened to either end) were two red, battery-powered, 5-LED bicycle lights for illumination; these red lights appeared to have little effect on beetle activity. We then covered the container with a glass pane taken from a picture frame. Trials ran from 18:00 until 07:00 the following day (13 h total), and were subject to natural light at a window, but controlled temperatures (21–24 °C). Sunset times ranged from 20:18 to 19:33; sunrise ranged from 06:04 to 06:35. We ran trials 10 August–6 September 2018; trials attempted after 6 September were abandoned because *N. orbicollis* ceased preparing carcasses for breeding. We recorded behaviors and interactions through the night using a Sony Handycam digital video recorder (DCR-SR100; Sony Corporation, Tokyo, Japan), with the "nightshot" option that allowed us to record video at night using the red lights. We covered the "nightshot" red light source of the video camera because it reflected off the glass, obscuring the view of the beetles, and instead relied on the red bicycle lights installed under the glass. After each trial, we disposed of the soil and carcass, washed the container with warm water, and introduced fresh soil. In total, we conducted 14 trials (7 male-male, 7 female-female).

**Table 1 Sex and measurements of beetles used in experiments.** Beetle sex and measurements of mass, elytra length, and pronotum width for beetles of each trial.

| Trial start date | Sex (same for both species) | Mass of *orbicollis* (g) | Mass of *tomentosus* (g) | Elytra length of *orbicollis* (mm) | Elytra length of *tomentosus* (mm) | Pronotum width of *orbicollis* (mm) | Pronotum width of *tomentosus* (mm) |
|---|---|---|---|---|---|---|---|
| Aug. 10 | female | 0.438 | 0.252 | 11.79 | 8.49 | 6.61 | 5.25 |
| Aug. 12 | male | 0.393 | 0.297 | 10.91 | 8.47 | 6.15 | 5.28 |
| Aug. 13 | male | 0.453 | 0.200 | 10.96 | 7.10 | 6.23 | 4.64 |
| Aug. 15 | male | 0.325 | 0.547 | 9.69 | 9.96 | 5.68 | 6.88 |
| Aug. 22 | female | 0.455 | 0.219 | 10.69 | 7.41 | 6.45 | 4.58 |
| Aug. 23 | female | 0.683 | 0.243 | 12.78 | 7.68 | 7.34 | 4.88 |
| Aug. 24 | male | 0.452 | 0.286 | 11.24 | 8.09 | 6.75 | 5.48 |
| Aug. 25 | female | 0.550 | 0.174 | 11.27 | 7.38 | 6.67 | 5.09 |
| Aug. 27 | male | 0.483 | 0.271 | 10.93 | 7.88 | 6.75 | 5.27 |
| Aug. 28 | female | 0.464 | 0.252 | 10.35 | 7.42 | 6.35 | 4.86 |
| Sep. 3 | male | 0.246 | 0.294 | 8.81 | 8.12 | 5.09 | 5.31 |
| Sep. 4 | female | 0.675 | 0.359 | 11.16 | 8.64 | 7.06 | 5.54 |
| Sep. 5 | male | 0.341 | 0.161 | 9.85 | 6.83 | 5.86 | 4.40 |
| Sep. 6 | female | 0.397 | 0.336 | 10.44 | 8.60 | 5.88 | 5.40 |

## Body size

After each trial, the individual beetles were frozen within sealed plastic bags for later measurement. Within a few weeks of completing the trials, we weighed each beetle to an accuracy of 0.001 grams using a GEM20 Smart Weigh jewelry scale (Smart Weigh Packaging Machinery Co., Ltd., Zhongshan City, Guangdong Province, China) and measured the width of the pronotum and length of the elytra to the nearest 0.01 mm using digital calipers (Table 1). To control for variation in the scale measurements, we also weighed a known 10 g standard and calculated beetle mass as: (beetle mass (g)) = (uncorrected weight of beetle (g)) × {(10g)/(weight of the 10g standard (g))}. We did not weigh beetles prior to the experiments because we wanted to minimize handling effects on their behaviors at the beginning of our experiments.

We preferentially used mass as an index of body size in our statistical analyses because elytra length and pronotum width may vary across species independent of size (e.g., as a result of adaptation to different ecological strategies), mass is directly important in some aspects of aggressive contests (e.g., inertia), and mass has been a standard estimate of size in previous work (*Peters, 1983*). Previous work also suggests that mass and elytra length are highly correlated within our two focal species at our study site (*Collard et al., 2020*). Nonetheless, we repeated our analyses with elytra length and pronotum width as indices of body size to ensure that results were similar (see "Statistical Analyses" below).

## Categorization of interactions

After conducting our experiments, we reviewed the video and categorized all interactions between our two focal species over the 13 h of each experiment. We first defined an

interaction between beetles as any occasion when two individuals came within 1 cm of each other. We used 1 cm as a cutoff point for interactions because our previous observations suggested that two individual beetles will be aware of each other's presence at this distance. Beetles may also be aware of each other at distances greater than 1 cm (e.g., through olfaction or stridulations), and thus our definition of an interaction may underestimate the overall number of interactions among beetles. Nonetheless, we had difficulty determining if an interaction was occurring at distances greater than 1 cm, so we excluded these from this study. For each interaction, we defined the initiator of the interaction as the individual that moved towards the other, such that the two individuals came within 1 cm of each other. We did not designate an initiator in cases where both individuals moved towards each other, bringing them <1 cm apart.

We separated interactions between beetles into four categories: (1) physical interactions and chases, (2) avoidance interactions, (3) neutral interactions, and (4) other interactions. Physical interactions and chases were defined as interactions that involved grappling, biting, chasing, and digging after a buried beetle. We further noted whether the outcome of these interactions was symmetric (no winner or loser in the interaction) or asymmetric (one beetle won and the other lost the interaction). A beetle that won an interaction retained its position, whereas a beetle that lost an interaction retreated (moved away) from the winner or carcass (no beetles were physically injured in any of our trials, thus all beetles that lost an interaction were able to retreat).

Avoidance interactions included interactions where we observed evidence for a dominant and subordinate, despite there being no physical contact or overt chasing. Examples include cases where one individual actively avoided the other individual (e.g., changing direction or approach) after coming within 1 cm of each other. Within avoidance interactions, we defined the subordinate as the individual moving away from the resource (the mouse).

Neutral interactions included all instances where an individual moved to <1 cm of the other without any sign of aggression or negative influence on each other's behaviors (e.g., no alteration of their path of movement or avoidance of the other beetle). Examples of neutral interactions include: the two beetles passed by each other without changing their direction, both beetles moved away from the mouse without avoidance or aggression, both beetles fed on the mouse without aggression, and one beetle passed over the other beetle (buried under the soil) without digging after or attacking it.

Some interactions were difficult for us to categorize. In some cases, interactions began as one category (e.g., neutral interactions) but changed over time into another category (e.g., physical interactions and chases). In these cases, we considered the sequential aggressive and non-aggressive interactions as distinct, and considered the initiator of the second interaction as the individual that closed the remaining distance between the two beetles. In other cases, both individuals were under the mouse simultaneously, and thus, we could not observe the occurrence or details of any interactions; in these cases, we recorded no interactions, or "na" for any component of an interaction that could not be observed (e.g., the initiating species). We designated "other interactions" as interactions where we could not view the details of the interaction (e.g., physical contact

vs avoidance), but nonetheless could tell that an interaction took place, and in some cases, the outcome of the interaction (e.g., one beetle retreated from the other).

## Assessment of behavioral dominance

Within each trial ($n = 14$), we designated the behaviorally dominant individual as the individual who won the most interactions with asymmetric outcomes (i.e., a winner and a loser), and the subordinate as the other individual (following *Drews (1993)*). Dominance designations based only on physical interactions and chases (i.e., avoidance and other interactions were excluded) were identical to dominance designations based on all interactions in 12/14 trials.

## First species to arrive at the carcass

For each of our 14 trials, we recorded the first beetle to physically touch the mouse as the first species to arrive at the carcass. This measure allowed us to test for priority effects on the outcomes of aggressive contests for the carcass on a time scale of minutes to hours, but did not address the potential for priority effects occurring on longer time scales (such as over days) that could also be important in nature.

## Behavioral dominance and distance to carcass

Competition experiments cannot always record the outcomes of individual fights, and thus we measured the distance of each beetle to the mouse carcass (cm) at the end of each trial (07:00) for future use as a proxy for behavioral dominance. Previous observations suggest that behaviorally dominant beetles will usually stay in close proximity to the carcass, while subordinate species are displaced further away (*Wilson, Knollenburg & Fudge, 1984*). We tested this hypothesis across the 14 trials in our experiment, predicting that the behaviorally dominant beetle (i.e., the beetle that had won most interactions) would be closer in proximity to the carcass than the subordinate, and thus the proximity to the carcass at the end of the trials would provide a useful estimate of behavioral dominance in competitive experiments.

## Statistical analyses

We performed all of our plotting and analyses in R (version 3.5.2; *R Development Core Team, 2018*).

We first tested among our alternative hypotheses using a binomial generalized linear model, with the proportion of asymmetric dominance interactions (i.e., physical interactions and chases, avoidance interactions, and other interactions with an asymmetric outcome; Table 2) won by each species as the response variable. Specifically, we combined the number of interactions won by *N. orbicollis*, and the number of interactions won by *N. tomentosus*, into one response variable (using the *cbind* command in R; *Crawley, 2013*), with the relative mass of the two species (ln (mass *N. orbicollis* in g/mass *N. tomentosus* in g)) and the first species to the carcass as the predictor variables in a saturated model (i.e., all interactions included). This model was overdispersed, so we corrected the standard errors using a quasibinomial generalized linear model. We tested the hypothesis that species identity consistently predicted interaction outcomes using the

**Table 2 Frequency of interactions among beetles, and the order, time and distance to carcass in each trial.** See "Materials and Methods: Categorization of Interactions" for definitions of interaction categories. For interactions with an asymmetric outcome (i.e., winner, loser), brackets denote the winner of the interaction (*orbicollis* winner, *tomentosus* winner). The species initiating asymmetric interactions is designated as: *orbicollis* initiated, *tomentosus* initiated; these cases do not sum to the total number of asymmetric interactions because a few interactions involved both individuals moving toward each other, and thus did not have a clear initiator.

| Trial start date | Physical interactions, chases[1] | Avoidance interactions[1] | Neutral interactions | Other interactions[1] | Total number of interactions[1] | Initiator of asymmetric interactions[2] | First to carcass | Time to carcass (min)[2,3] | Distance to carcass in morning (cm)[2] |
|---|---|---|---|---|---|---|---|---|---|
| Aug. 10 | 4 (4,0) | 2 (2,0) | 99 | 2 (2,0) | 107 (8,0) | 7,1 | *tomentosus* | 1.2, 0 | 4.0, 10.0 |
| Aug. 12 | 9 (9,0) | 5 (5,0) | 72 | 1 | 87 (14,0) | 13,1 | *tomentosus* | 155.6, 0 | 0.0, 15.0 |
| Aug. 13 | 3 (3,0) | 5 (5,0) | 74 | 1 (1,0) | 83 (9,0) | 3,5 | *orbicollis* | 4.0, 55.3 | 8.5, 10.0 |
| Aug. 15 | 30 (3,19) | 19 (4,15) | 25 | 0 | 74 (7,34) | 23,18 | *orbicollis* | 4.3, 6.9 | 4.5, 0.0 |
| Aug. 22 | 34 (16,10) | 55 (34,21) | 91 | 4 | 184 (50,31) | 43,38 | *tomentosus* | 120.2, 38.3 | 3.5, 0.0 |
| Aug. 23 | 10 (6,0) | 5 (4,1) | 49 | 10 (7,0) | 74 (17,1) | 13,5 | *orbicollis* | 0, 6.2 | 0.0, 6.5 |
| Aug. 24 | 10 (10,0) | 6 (5,1) | 116 | 6 | 138 (15,1) | 13,2 | *orbicollis* | 1.3, 24.6 | 0.0, 5.0 |
| Aug. 25 | 8 (8,0) | 3 (3,0) | 13 | 1 (1,0) | 25 (12,0) | 12,0 | *tomentosus* | 3.7, 0 | 3.0, 9.5 |
| Aug. 27 | 4 (4,0) | 14 (13,1) | 126 | 1 | 145 (17,1) | 7,11 | *tomentosus* | 188.1, 150.9 | 1.5, 10.0 |
| Aug. 28 | 17 (4,5) | 54 (31,23) | 70 | 3 | 144 (35,28) | 34,29 | *tomentosus* | 214.2, 159.0 | 0.0, 9.0 |
| Sep. 3 | 0 | 0 | 88 | 2 | 90 (0,0) | 0,0 | *orbicollis* | 19.8, 45.9 | 0.0, 17.5 |
| Sep. 4 | 6 (1,5) | 26 (15,11) | 134 | 0 | 166 (16,16) | 13,19 | *tomentosus* | 71.6, 70.4 | 0.0, 0.0 |
| Sep. 5 | 24 (11,3) | 16 (8,8) | 23 | 0 | 63 (19,11) | 12,17 | *orbicollis* | 0.4, 1.2 | 0.0, 0.0 |
| Sep. 6 | 9 (9,0) | 1 (1,0) | 88 | 0 | 98 (10,0) | 10,0 | *tomentosus* | 193.2, 0 | 0.0, 4.0 |

**Notes:**

[1] Number of asymmetric interactions won by: (*orbicollis*, *tomentosus*).

[2] Values for: *orbicollis*, *tomentosus*.

[3] Values indicate the time since the beginning of video recording; zero indicates a beetle was already in contact with the carcass at the beginning of video recording

intercept of the model, which provides a test of asymmetry in the interaction outcomes when the difference in mass between individuals is zero. We removed one trial that had no interactions between the two species during the 13 h of observation.

We ran models with all possible combinations of predictor variables and compared their performances using quasi Akaike information criterion values (QAIC) and the *dredge* command in the R package *MuMIn* (*Bartoń, 2019*). We calculated QAIC values because AIC values are not available for quasibinomial generalized linear models; we calculated QAIC values following *Bolker (2014)*. We identified the best-performing model as the model with the lowest QAIC value. We summarize the results of the best-performing model in the Results section, and present model performance results in Table 3. In addition to running models with mass as our proxy for body size, we repeated our analyses with both elytra length and pronotum width as estimates of body size.

## RESULTS

### Interactions among the species

The total number of interactions, and their distribution among the different categories, varied between trials, with most interactions being neutral (Table 2). Most individuals engaged in both physical interactions and chases and avoidance interactions, with no clear patterns between trials (Table 2). The majority of physical interactions and chases were

**Table 3 Comparison of model performance for quasibinomial generalized linear models.** Different combinations of predictors were evaluated for their ability to explain behavioral dominance among *Nicrophorus* beetles. The response variable was the proportion of aggressive and dominance interactions won by *N. orbicollis* or *N. tomentosus* ($n$ = 13 trials).

| Intercept[1] | First species to carcass | Relative mass | First species to carcass: Relative mass | df | logLik | QAIC | ΔQAIC | weight |
|---|---|---|---|---|---|---|---|---|
| −0.1953 | —[2] | 1.599 | — | 2 | −54.182 | 28.1 | 0 | 0.691 |
| 0.6215 | — | — | — | 1 | −71.784 | 31 | 2.86 | 0.165 |
| −0.09265 | +[3] | 1.697 | — | 3 | −53.882 | 32.4 | 4.23 | 0.084 |
| 0.3545 | + | — | — | 2 | −70.337 | 34 | 5.81 | 0.038 |
| −0.2546 | + | 2.344 | + | 4 | −45.766 | 35 | 6.88 | 0.022 |

**Notes:**
[1] Numbers are intercept estimates or effect sizes (for predictor variables).
[2] Long dash (—) indicates that the predictor variable was absent from the model.
[3] +Indicates that a factor or interaction term was included in the model.

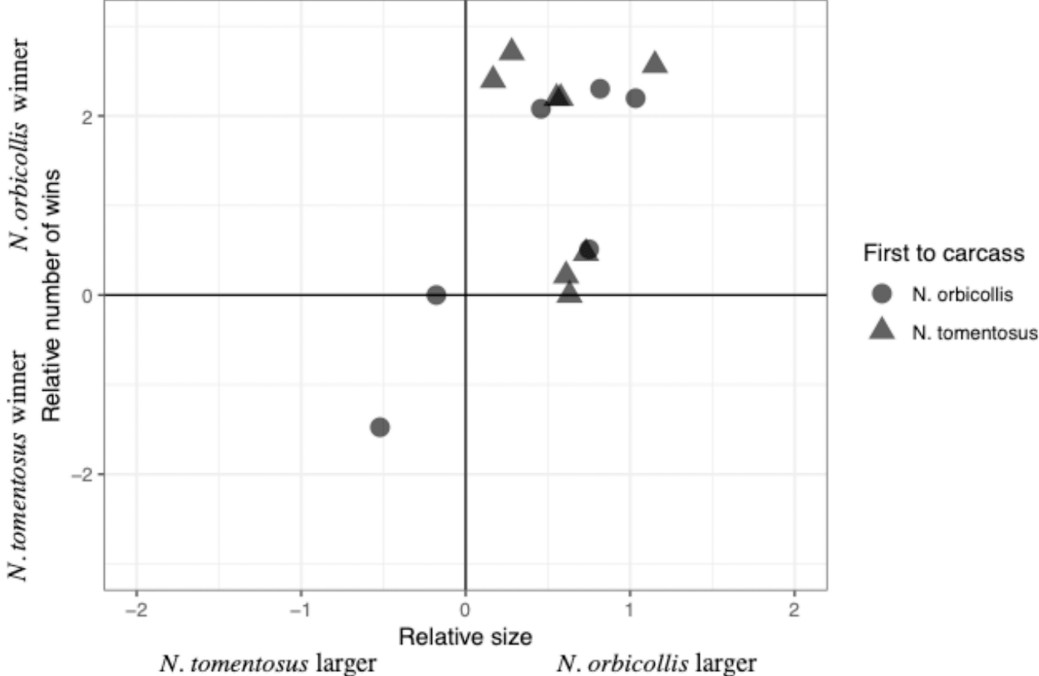

**Figure 1 Comparison of the relative interaction wins to relative beetle size.** Comparison of the relative number of wins {ln((*N. orbicollis* wins+1)/(*N. tomentosus* wins+1))} to relative beetle size (ln(mass *N. orbicollis*/mass *N. tomentosus*)), with shape indicating which species was first to the carcass across all 14 trials.

asymmetric, with a clear loser retreating from its position. These interactions usually consisted of grappling or chasing, with some other less common interactions, including digging after the other beetle. No beetles were visibly injured due to an aggressive interaction. Interaction time varied across experiments with an average of 11.85 +/− 40.65s (SD; $n$ = 6), with neutral interactions lasting longer than other interactions. Both species initiated interactions with asymmetric outcomes (i.e., a winner and loser), despite *N. orbicollis* usually winning (Fig. 1; Table 2). *N. orbicollis* initiated more interactions with

an asymmetric outcome than *N. tomentosus* in 9 of the 14 trials (Table 1); *N. orbicollis* initiated an average of 14.50 +/− 11.64 (SD) interactions, while *N. tomentosus* initiated an average of 10.43 +/− 12.13 (SD) interactions (Table 2).

### Factors predicting behavioral dominance

Larger relative body size was the best predictor of behavioral dominance between *N. orbicollis* and *N. tomentosus*. Our best-performing model (Table 3) included only relative body size as a predictor variable (quasibinomial linear model, estimate = 1.60 +/− 0.69SE, $t = 2.3$, $p = 0.04$, $n = 13$; Fig. 1). The intercept of the model—our test of a species effect—was not significantly different from zero (estimate = −0.20 +/− 0.46SE, $t = −0.43$, $p = 0.68$), while the first species to arrive at the carcass (i.e., priority effect) was not retained in any top model (Table 3). The first beetle to the carcass arrived between zero (i.e., the beetle went almost immediately to the carcass, and was on the carcass by the time we started recording) and 159.0 min (mean = 32.1 min) after video recording began; the second beetle arrived 1.2–193.2 min (mean = 45.7 min) after the first (Table 2).

Generalized linear model results were similar if we used elytra length or pronotum width as indices of body size (elytra length: best-performing model included only elytra length as a predictor; elytra length estimate = 5.83 +/− 2.16SE, $t = 2.7$, $p = 0.02$, intercept = −1.10 +/− 0.70SE, $t = −1.6$, $p = 0.14$; pronotum width: best-performing model included only pronotum width as a predictor; pronotum width estimate = 3.73 +/− 1.78SE, $t = 2.1$, $p = 0.06$, intercept = −0.19 +/− 0.49SE, $t = −0.39$, $p = 0.70$).

### Behavioral dominance and distance to carcass

When the two species differed in their distance to the carcass (measured at 07:00, just after the competition experiment), the closest individual to the carcass also won the most interactions in 10 of 12 trials (83%; binomial test on accuracy, $p = 0.039$). In only one trial was the subordinate individual (i.e., the individual that lost most interactions) closer to the carcass than the dominant individual (in the other trial, the two beetles won the same number of interactions). In the remaining two of the 14 trials, the two beetles were equidistant from the carcass at the end of the experiment; in one of these trials, the two beetles also won equal numbers of interactions, and thus the distance to the carcass accurately described the lack of a dominance relationship. In one of the 14 trials, the two beetles were not observed to interact during the 13-h experiment (Table 2); however, the larger *N. orbicollis* was closer to the carcass at the end of the experiment.

### DISCUSSION

Relative body size best predicted the outcome of dominance interactions between *Nicrophorus orbicollis* and *N. tomentosus*. Larger beetles won the majority of interactions, regardless of species, and as the relative size difference between the beetles increased, so did the difference in the number of wins between species (Fig. 1). We found no support for the alternative hypotheses that one species consistently dominates regardless of size, or that the first individual to the carcass is more likely to win subsequent interactions (Table 3). Our results cannot rule out an influence of species identity or order of

arrival (priority effects) on the outcomes of aggressive interactions—more extensive experiments that controlled for other factors (e.g., matched opponents for size) could still find effects. But our results nonetheless suggest that relative size exerts a larger effect on the outcome of aggressive contests in these species relative to either species identity or priority effects.

Aggressive interactions between the beetles were common and involved behaviors where large body size may have provided advantages. When physically interacting, the beetles engaged in grappling (see *Pukowski (1933)*), chasing, and digging in the soil after submerged competitors. These interactions could benefit from the increased strength, stability, and energy reserves conferred by larger and heavier bodies (*Enders, Schüle & Henschel, 1998*; *Safryn & Scott, 2000*; *Davis, Attarha & Piefke, 2013*). Likewise, body size may positively covary with suites of other traits involved in aggressive interactions, such as legs and mandibles (*Pukowski, 1933*; *Enders, Schüle & Henschel, 1998*; *Lailvaux et al., 2005*; *Benowitz, Brodie & Formica, 2012*). For example, larger beetles may have longer legs and more robust leg musculature, possibly resulting in faster running and digging speeds during chases, and greater reach, grip strength, and pushing force when grappling (*Forsythe, 1983*; *Enders, Schüle & Henschel, 1998*; *Benowitz, Brodie & Formica, 2012*).

Body size could also act as a signal of competitive ability prior to aggressive interactions. Interactions between the beetles were often initiated when one beetle had possession of the carcass and the other approached. Once within 1 cm of each other, the initiating beetle would either turn around and retreat from the mouse or alter their course to move closer to the other beetle, frequently resulting in physical interactions. Body size could serve as a visual signal of competitive ability that the beetles assess when close to one another and may provide information that influences whether the initiator retreats to safety or contests the carcass. In particular, differences in pronotum width could decide dominance without escalating into fighting (*Safryn & Scott, 2000*). Other signals may also be important for signaling competitive ability, as burying beetles also use olfactory and acoustic cues to communicate (*Pukowski, 1933*; *Chemnitz et al., 2015*), but we were unable to evaluate these signals in this study.

Our findings are consistent with previous studies that found large body size to be the best predictor of behavioral dominance in competition experiments between *Nicrophorus* species (*Pukowski, 1933*; *Easton, 1979*; *Wilson, Knollenburg & Fudge, 1984*; *Otronen, 1988*; *Trumbo, 1990a*; *Scott, 1998*; *Ye, 2019*). However, unlike the findings of *Otronen (1988)*, which found that some species (*N. vespilloides*, *N. investigator*, and *N. vespillo*) were behaviorally dominant even when the same size as their competitors, we found no effect of species identity influencing the outcome of interactions. In our experiments, the larger beetle won more interactions and was found to be behaviorally dominant more often regardless of the identities of themselves or their competitor. Our study also supports previous findings that the first species to reach a carcass does not always retain it or win in subsequent interactions (*Bartlett & Ashworth, 1988*; *Trumbo, 1990a*; *Wilson, Knollenburg & Fudge, 1984*; see *Otronen, 1988* for examples of priority effects within *N. vespilloides*).

*Nicrophorus tomentosus* may be able to coexist and thrive with larger, more dominant species of *Nicrophorus* by tolerating adverse environmental conditions. *N. orbicollis* is on average larger than *N. tomentosus* at our study sites (*Collard et al., 2020*), and thus *N. orbicollis* may competitively exclude most *N. tomentosus* from carcasses when the two species overlap their breeding seasons (in late July and August; *Wettlaufer, 2019*). *N. tomentosus*, however, can breed later in the season, after other co-occurring *Nicrophorus* have ceased breeding. The ability of *N. tomentosus* to overwinter as prepupae, and the consequential shorter development time before winter diapause, may allow *N. tomentosus* unique access to the late breeding season at our site, providing a period free from competition with larger breeding *Nicrophorus* (*Anderson, 1982*; *Wilson, Knollenburg & Fudge, 1984*; *Trumbo, 1990b*; *Wettlaufer, 2019*).

The question remains: Does overwintering as prepupae compromise the ability of *N. tomentosus* to reach larger adult sizes, thus creating a trade-off between competitive ability (determined by larger body size) and the ability to breed later in the season? We are currently unaware of any evidence to support such a trade-off and we know little overall about the costs of prolonged prepupal diapause in *N. tomentosus*. Alternatively, breeding in late summer and fall may compromise the ability of *N. tomentosus* to develop larger adult body size through processes unrelated to wintering as prepupae (e.g., greater competition from flies (Diptera) for carcasses; *Scott, 1994*). Likewise, other selective pressures associated with *N. tomentosus* ecology, such as their tendency to fly during the day (*Wilson, Knollenburg & Fudge, 1984*; *Anderson & Peck, 1985*), could also constrain their ability to evolve large sizes typical of *N. orbicollis* (e.g., risk of overheating during flight with increased size). Small-sized *Nicrophorus* (*N. defodiens*) are able to use carcasses more efficiently (producing more larvae per gram of carcass) compared with larger species (*N. orbicollis*) (*Trumbo, 1990a*); such benefits may be favored at different times of the year or under different environmental conditions. We look forward to future experiments that can test among these alternative ideas for possible trade-offs between behavioral dominance through large size, and the ability of *N. tomentosus* to reproduce late in the breeding season.

## ACKNOWLEDGEMENTS

Thanks to Dr. Frances Bonier for comments on the study and earlier versions of the manuscript, and anonymous reviewers for helpful feedback. Thanks to Sarah Mayo, Mia Marcellus, Michelle Cohen and Emily Elliot for their help in data collection. Thank you to everyone else in the Bonier and Martin labs for support.

### Funding

This work was supported by the Natural Sciences and Engineering Research Council of Canada Discovery Grant RGPIN/355519-2013. The funders had no role in study design, data collection and analysis, decision to publish, or preparation of the manuscript.

## Grant Disclosures

The following grant information was disclosed by the authors:
Natural Sciences and Engineering Research Council of Canada: RGPIN/355519-2013.

## Competing Interests

The authors declare that they have no competing interests.

## Author Contributions

- Scott D. Schrempf conceived and designed the experiments, performed the experiments, analyzed the data, prepared figures and/or tables, authored or reviewed drafts of the paper, and approved the final draft.
- Kevin W. Burke conceived and designed the experiments, authored or reviewed drafts of the paper, and approved the final draft.
- Jillian D. Wettlaufer conceived and designed the experiments, authored or reviewed drafts of the paper, and approved the final draft.
- Paul R. Martin conceived and designed the experiments, performed the experiments, analyzed the data, prepared figures and/or tables, authored or reviewed drafts of the paper, and approved the final draft.

## Data Availability

Raw interaction categorization data used for statistical analysis is available as a Supplemental File.

The code file used to statistically analyze data presented in table one and two is available at GitHub: https://github.com/sschrempf/Supplementary-Code-S1.

## Supplemental Information

Supplemental information for this article can be found online at http://dx.doi.org/10.7717/peerj.10797#supplemental-information.

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
