# Peer review of "Behavioral dominance interactions between Nicrophorus orbicollis and N. tomentosus burying beetles (Coleoptera: Silphidae)"

_PeerJ, doi:10.7717/peerj.10797_

## Round 0.1 · original submission · Major Revisions

Dear Dr. Schrempf and colleagues:

Thanks for submitting your manuscript to PeerJ. I have now received two independent reviews of your work, and as you will see, the reviewers raised some concerns about the research. Despite this, these reviewers are optimistic about your work and the potential impact it will have on research studying the behavioral ecology of Nicrophorus burying beetles. Thus, I encourage you to revise your manuscript, accordingly, taking into account all of the concerns raised by both reviewers.

While the concerns of the reviewers are relatively minor, this is a major revision to ensure that the original reviewers have a chance to evaluate your responses to their concerns. There are many suggestions, which I am sure will greatly improve your manuscript once addressed.

Please defend your rationale for using body mass over other variables as a competition metric, as well as your approach for determining body mass. Please also be clear with explaining terms that are specific for your field; i.e., “carcass preparation” would be unclear for general readers. Please also address “larger is not necessarily better” (R2).

Therefore, I am recommending that you revise your manuscript, accordingly, taking into account all of the issues raised by the reviewers.

Good luck with your revision,

-joe

Reviewer 1 ·

Basic reporting

see below

Experimental design

see below

Validity of the findings

see below

Additional comments

Manuscript #53206 examines contest competition between individuals of two species of burying beetles that have a competitive size asymmetry. The experimental design is straightforward and easy to interpret and the English is understandable. The literature was covered very nicely. Most of my comments are relatively minor.

A primary question I had was why use body mass for the size variable rather than another body size measure. The methods indicate that length and width were measured but these measures are never mentioned again (if not used, why mention them at all)? Body mass in burying beetles is known to be quite variable as a beetle under short-term starvation can increase body mass by 10% during one active period of feeding. Length and width measures are not so variable. Was body mass used because width of a body part, such as the the pronotum, may not accurately indicate relative size across different species with different body proportions? If so, this could be mentioned. I also thought it unusual to weigh body mass after the experiment. Dominance interactions can influence access to the carcass, and hence alter feeding and weight gain that might have occurred after dominance was determined.

There is some speculation in the last paragraph of the manuscript on what affects body size evolution in burying beetles. Competition is a natural explanation. Two large species, however, N. marginatus and N. carolinus, are found in habitats where there is almost no interspecific competition and perhaps limited intraspecific competition (open habitats generally do not have as large of beetle populations as wooded habitats). Both species, however, bury carcasses very deeply and larger body size may be important to achieve this (whether they bury deeply to avoid competition or to reach a depth where there are not elevated daytime temperatures is unclear). This is another possible trade-off. A primary trade-off that I think should be mentioned is that smaller species can raise greater numbers of young for a given carcass size.

The introduction focuses on the trade-off of size and environmental tolerance, but I do not see how breeding at the end of the season constrains body mass in this group (is the end of the season considered a poor nutritional environment?). Body size is determined by feeding as a larva so I do not see that fly competition would affect body since beetles breeding on a small carcass generally do so only if fly eggs and larvae are killed. Body size is determined by the parental decision on how many young to raise on a given sized resource so this trait should be flexible and selected by factors such as competition, digging ability, reserves needed to survive overwintering, flight abilities, etc. Do other species of burying beetles that breed late in the year and overwinter as prepupae (Europe, Asia) have smaller body size, on average? If so, this may be a clue.

I agree that the smaller N. tomentosus can avoid some competition with orbicollis by breeding later in the season but, as I see it, the constraint here is that you can only breed for 3-4 weeks instead of 10 weeks.

L. 62-64. Of the 7 species of Nicrophorus in Ontario, N. pustulatus has not been confirmed as a breeder on small vertebrate carcasses in the field, as far as I know.

L. 103/ L. 336/ L. 340. In some cases, the overwintering stage of N. tomentosus is given as “larva” whereas in others it is “prepupa”. I suggest a consistent terminology (I prefer prepupa as larvae may imply the feeding stage).

L. 105. Overwintering as a larva is not unique to N. tomentosus. In North America, N. investigator and N. hybridus do so, as well as additional species in Eurasia. There is also at least one species that facultatively overwinters in a pre-adult stage.

L. 142. Why are the sunset times given in reverse chronological order?

L. 156. I have not seen a scale correction for measurements of animal mass in a methods section. Were different scales used? With the considerable environmental variation in body mass (see above), measuring to 0.001 g seems unnecessary.

L.235. Is a “two-vector variable” common terminology? Does this just mean that both aggressive and non-aggressive interactions were used in separate analyses of won/loss? Or were the two measures combined to determine won/loss?

The explanatory variable, species identity, might be better tested when beetles of different species are matched for size. With no cases of similar-sized opponents, this variable was likely difficult to analyze, with limited statistical power. The results do confirm, however, that this variable is not of major importance and that size is very important.

The results also confirm that arrival time (at least over periods of hours) is not an important determinant of contest outcome. Once larvae have been on the carcass for a couple of days, smaller residents can sometimes keep larger intruders off the carcass, presumably because of differences in motivation of resident and intruder at that stage.

With a sample of 13, there is likely not enough data to look at the effects of other variables. But was there any indication that sex is an important variable? Were the findings similar for male pairs and female pairs? Did male-male contests have fewer overall aggressive interactions?

Was there an appearance of a date effect? If N. orbicollis stopped breeding on September 7th, perhaps some N. orbicollis during the latter part of the study were less motivated to contest for a carcass.

Reviewer 2 ·

Basic reporting

The authors use clear and unambiguous, professional English. The article is professionaly structured. I suggest to add some background information and references (see general comments).

Experimental design

See general comments

Validity of the findings

See general comments

Additional comments

The authors present an interesting study that investigates behavioural dominance interactions between two burying beetle species co-occurring in the same habitat. Burying beetles reproduce on dead vertebrates and have been shown to compete intra- as well as interspecifically for the limited resource. The study video-recorded behavioural interactions of the two species for 13 hours and tested whether body size, species identity or time of arrival are the best predictor of behavioural dominance. The authors found that large body size determined the outcome, whereas species identity or the order of arrival did not predict behavioural dominance.

The manuscript is clearly and well written. Despite the wealth of knowledge about the behaviour of burying beetles, our understanding of interspecific interactions is still little and therefore, the current study is interesting and timely. You will see below that I miss some background information and references and I also have some questions and comments to the experimental design.

1. I suggest mentioning that burying beetles engage in pre- and post-hatching care. This is relevant, as the beetles continue to defend the carcass until the offspring disperse. There is a nice review of Eggert and Müller 1997 about the natural history and behaviour of burying beetles (“Biparental care and social evolution in burying beetles: Lessons from the larder”).

2. Burying beetles do usually try to monopolize a carcass, however, they also have been shown to engage in cooperative breeding and inta- and interspecific brood parasitism. Especially interspecific brood parasitism should be mentioned in the manuscript (e.g. Trumbo 1994). N. tomentosus might have the ability to successfully parasite N.orbicollis broods.

3. Line 74-77: not only the seasonal activity pattern, but also the daily activity pattern can explain species coexistence. As a matter of fact, since N. tomentosus is diurnal and N. orbicollis nocturnal, they search for carcasses at different times and might not regularly compete for carrion resources.

4. Line 108: readers might not necessarily understand what is meant by carcass preparation; furthermore, neither Otronen nor Pukowski have studied N. orbicollis or N. tomentosus. There are many other studies you could cite. E.g. studies of Trumbo, Scott or Creighton. Steiger et al 2012 and Pettinger et al 2011 (both published in Behavioral Ecology) also examined dominance behaviour in N. orbicollis. In fact, the first study found that body size was a very good predictor of dominance.

5. Line 126: Some of the beetles were kept for more than a week. Since age might have an effect on the fighting ability (due to senescence) or on the investment in fights (due to terminal investment), it should be mentioned how long individuals of both species have been housed outdoors before the start of the experiment. Possible species difference in the time they have been maintained might have affected the results. A similar problem arises due to the fact that they have been fed every 7 days. The nutritional state likely affects fighting abilities (see e.g. Richardson et al. 2020). Beetles that were kept more than a week were fed ground beef. But what about beetles that were kept less than a week? A beetle caught in field that had been kept six days without any nutrition might not have the strength to engage in fights (or vice versa might fight vigorously for the possession of a carcass).

6. Line 140: why was the trials run overnight and not during the day? Since N. orbicollis is nocturnal and N. tomentosus diurnal, this might have biased the outcome of the interactions.

7. Line 151: what is meant by 14 trials? If 7 males and 7 females were involved, there were only 7 trials? I thought pairs of beetles consisted of the same sex? But then, the number of males and females must be even. Was there any effect of sex? E.g. did one sex fight less? Experiments by Dressel (1987), Bartlett (1988) and Bartlett and Ashworth (1988) have shown that conspecific males do not fight vigorously on carcasses unless a female is present.

8. Line 178 what is meant by “retained [..] its physical condition”?

9. Perhaps I missed something. For body size you used body weight, but not pronotum size?

10. Line 214-217: Priority effects: since both beetles were introduced at the same time, the time difference in carcass discovery cannot be large. How much was the time difference?

11. General comment to the methods: I was wondering why you did not look at the reproductive outcome (in the case of females). It would have been nice to see whether interactions led to a complete exclusion of another individual from the carcass or whether individuals of both species produced eggs and had offspring on the same cadaver.

12. Line 316: there is a lot of literature about chemical and acoustic communication in burying beetles (e.g. Chemnitz et al. 2015 Proc R Soc; Keppner et al. 2017 J Chem Ecol; Steiger et al. 2007 Proc R Soc; Engel et al. 2016 Nature Com; Engel et al. 2019 Chemoecology; Hall et al 2013 Annals Entomol Soc Am). Pukowski did not examined any potential olfactory cues.

13. Line 340-352: Be aware that larger is not always better. Smaller species have the advantage that they can use smaller carcasses for reproduction (e.g. N. vespilloides can reproduce on carcasses with a weight of 2g; N. orbicollis won’t be able to use such small resources). Even if they used carcasses of similar size, N. tomentosus might produce smaller, but a higher number of offspring. N. americanus is the largest species in North America, but endangered, which illustrates that a large body size is not necessarily the best strategy.

Reviewer 3 ·

Basic reporting

no comment

Experimental design

no comment

Validity of the findings

no comment

Additional comments

This article shows that body size is the key to the success in inter-specific competition and the use of carcasses. Also, the probability of winning the contest over carcasses is independent of who arrives at the carcass first. The article is simple and the conclusion is clear. Although I have read the article several times, (surprisingly) I don't have any suggestions for the author to revise.

---

## Round 0.2 · accepted · Accept

Dear Dr. Schrempf and colleagues:

Thanks for revising your manuscript based on the concerns raised by the reviewers. I now believe that your manuscript is suitable for publication. Congratulations! I look forward to seeing this work in print, and I anticipate it being an important resource for groups on research studying the behavioral ecology of Nicrophorus burying beetles. Thanks again for choosing PeerJ to publish such important work.

Best,

-joe

Reviewer 1 ·

Basic reporting

I have no further comments on the revised manuscript. The authors have addressed my concerns.

Experimental design

N/A

Validity of the findings

N/A

Additional comments

The revision was well done!